# TROPESS-CrIS CO single pixel vertical profiles: Intercomparisons with MOPITT and model simulations for 2020 US Western wildfires

Ming Luo[1], Helen M. Worden[2], Robert D. Field[3,4], Kostas Tsigaridis[3,5], and Gregory S. Elsaesser[3,4]

[1]Jet Propulsion Laboratory, California Institute of Technology, Pasadena, CA 91109, USA
5 [2]National Center for Atmospheric Research, 1850 Table Mesa Dr, Boulder, CO 80305, USA
[3]NASA Goddard Institute for Space Studies, 2880 Broadway, New York, NY 10025 USA
[4]Department of Applied Physics and Applied Mathematics, Columbia University, 2880 Broadway, New York, NY 10025 USA
[5]Center for Climate Systems Research, Columbia University, 2880 Broadway, New York, NY 10025 USA

10 *Correspondence to*: Ming Luo (Ming.Luo@jpl.nasa.gov)

**Abstract.** The new TROPESS (TRopospheric Ozone and its Precursors from Earth System Sounding) profile retrievals of Carbon Monoxide (CO) from the Cross-track Infrared Sounder (CrIS) are evaluated against Measurement of Pollution in the Troposphere (MOPITT) CO Version 9 data. Comparison results that were adjusted to common a priori constraints in the retrieval processes have improved agreement between the two data sets over direct comparisons. TROPESS-CrIS CO profiles are within 5% of MOPITT but have higher concentrations in the lower troposphere and lower concentrations in the upper troposphere.  For the intense Western US wildfire events in September 2020, we compare GISS climate model simulated CO fields to the two satellite CO observations. We show intermediate steps of the comparison process to illustrate the evaluation of model simulations by deriving the "retrieved" model CO profiles as they would be observed by the satellite. This includes the application of satellite Level 2 data along with their corresponding diagnostic operators provided in the TROPESS-CrIS and MOPITT products. The process allows a diagnosis of potential model improvements in modelling fire emissions and pollution transport.

## 1 Introduction

As a direct pollutant to Earth's atmosphere, Carbon Monoxide (CO) is a product of incomplete combustion which has a lifetime of weeks to months. CO has therefore been used as a tracer in atmospheric ozone-related photochemistry and pollution transport studies. High concentrations of CO in source regions can be seen above background concentrations from, for example, biomass burning, traffic and other fossil fuel combustion in polluted cities and industrial areas (Jacob, 1999). High-CO plumes are seen extending downwind to pollute nearby regions and sometimes circling the globe. For more than twenty years, retrievals of vertical CO profiles or total columns based on satellite measurements of CO absorption bands of 4.7 μm and 2.3 μm have been made available from several platforms, such as Measurement of Pollution in the Troposphere (MOPITT) (Drummond et al., 2009), Atmospheric Infrared Sounder (AIRS) (Aumann et al., 2003), Tropospheric Emission Spectrometer (TES) (Beer et al., 2001), Infrared Atmospheric Sounding Interferometer (IASI) (Clerbaux et al., 2009), Cross-track Infrared Sounder (CrIS) (Han and Revercomb et al., 2013), and TROPOspheric Monitoring Instrument (TROPOMI) (Veefkind et al., 2012). Also, some limited CO data also became available, e.g., from GOSAT (Noël et al., 2022) and GIIRS (Zeng et al., 2023) satellite observations. These satellite CO observations are valuable in tracking and quantifying pollutant emissions, horizontal gradients in pollution patterns, and the pollutant photochemical processes occurring as air moves (Clerbaux et al., 2002). From a long-term and global point of view, the satellite CO observations have been used to track the pollution-time trend in geophysical regions annually and seasonally (Worden et al., 2013; Buchholz et al., 2021). The satellite CO observations have also been used to evaluate parameter variations that drive model simulations of the atmospheric system (Field et al., 2015 & 2016; Buchholz et al., 2018).

In this paper, we focus on (1) satellite retrieved profile comparisons between the newly available CrIS CO from TRopospheric Ozone and its Precursors from Earth System Sounding (TROPESS) (Bowman, 2021, Worden et al., 2022), which uses the MUSES (Multi-SpEctra, Multi-SpEcies, Multi-Sensors of Retrievals of Trace Gases) algorithm (Fu et al., 2016) and the MOPITT V9 data (Deeter et al., 2022), and (2) comparisons of TROPESS-CrIS and MOPITT CO to the NASA Goddard Institute for Space Studies (GISS) model simulations for September 2020 wildfire events in the Western US. We present some details in understanding the a priori constraints used by different instrument retrieval algorithms and their influences in the final retrieval products. We illustrate how the measurement and retrieval characteristics provided in the data products should be used in data applications, such as model-evaluation processes aimed to improve some parameters important in model development.

## 2 Satellite CO observations: TROPESS-CrIS and MOPITT comparison

### 2.1 CO retrievals from two satellite observations

The observation configurations of CrIS instrument on NOAA S-NPP and NO-20 and the MOPITT instrument on NASA Terra satellites are listed in Table 1. The spectral sensitive ranges of both instruments cover Carbon Monoxide absorption bands in Thermal Infrared (TIR); MOPITT also covers the CO absorption band in Near-IR. For the CrIS and MOPITT comparison studies in this paper, we do not consider the MOPITT data products using NIR or TIR/NIR-combined CO retrievals.

**Table 1.** Observation Configurations for CrIS and MOPITT

|  | Launch Time | Orbit Nadir Local Time | Swath Width | Footprint Size at Nadir | Instrument Type and Spectral Range for CO Retrieval | Daily Num of Pixel Scan Observations |
|---|---|---|---|---|---|---|
| CrIS | S-NPP (Since Oct 2011) NOAA-20 (Since Nov 2017) | S-NPP 1:30 & 13:30 NOAA-20 2:20 & 14:20 | 2200 km | 14 X 14 km | Fourier Transform Spectrometer (TIR 2160-2200 cm$^{-1}$) | ~3.5 million |
| MOPITT | NASA-Terra (Since Dec 1999) | 10:30 & 22:30 | 650 km | 22 X 22 km | Gas Filter Correlation Radiometer (TIR 2140-2195 cm$^{-1}$) | 300K |

CrIS is an infrared Fourier transform spectrometer on-board NOAA Suomi- National Polar-orbiting Partnership (Suomi-NPP) and Joint Polar Satellite System-1 (JPSS-1 or NOAA-20) satellite operating since 2011 and 2017 respectively (https://www.jpss.noaa.gov/mission_and_instruments.html and https://www.star.nesdis.noaa.gov/jpss/CrIS.php). Its sun-synchronous orbits cover the entire globe with 16 days local footprint repeat time. Under the TROPESS project, the MUSES data processing system (Fu et al., 2016) inherited from the TES project is running forward in-time providing CrIS CO and other atmospheric gas retrievals at a reduced global sampling – one every 0.8 degrees latitude and longitude box. SNPP/CrIS makes measurements at local early afternoon (13:30) and after midnight hours. Each CrIS pixel, or field of view (FOV) is circular with a 14 km radius at nadir. The TROPESS-CrIS CO products use single pixel radiances with the MUSES algorithm (Fu et al., 2016, 2018, 2019) that applies an optimal estimation retrieval approach (Rodgers, 2000) with heritage from Aura/TES (Tropospheric Emission Spectrometer) Level 2 processing (Bowman et al., 2006). The TROPESS retrieval approach and TROPESS-CrIS CO products differ from other available CrIS CO products that combine 9 FOVs to obtain a single cloud-cleared radiance and corresponding retrieval of atmospheric parameters such as the NOAA Unique Combined Atmospheric Processing System (NUCAPS) (Gambacorta et al., 2013, 2014) and the Community Long-term Infrared Microwave Combined Atmospheric Product System (CLIMCAPS) (Smith and Barnet, 2020).MOPITT is a satellite gas-filter correlation radiometer (GFCR) instrument that has been operating on NASA Terra since 1999 with over 23 years of data (https://www2.acom.ucar.edu/mopitt). The on-board gas absorption cells and radiometers are used to derive CO concentrations in the atmosphere similar to a high spectral resolution spectrometer (Drummond et al., 2010). Terra orbits also repeat every 16

75    days, and MOPITT obtains global coverage in ~3 days. However, MOPITT observation local times are 10:30 and 22:30, different from that of SNPP/CrIS. The MOPITT FOV is 22km x 22km.

## 2.2 The comparisons between CO retrievals from the two satellite observations

Here we compare MOPITT and TROPESS-CrIS CO vertical profile retrievals from MOPITT V9T data (version 9, thermal infrared only) and the TROPESS Release 1.12 data. Both MOPITT and TROPESS-CrIS CO retrievals have been validated

against aircraft in-situ and other satellite measurements (George et al., 2009, 2015; Luo et al., 2007a & b; Deeter et al., 2019; Hegarty et al., 2022, Worden et al., 2022). Although the two data sets demonstrate general agreement in global distribution patterns in the lower, middle and upper troposphere, such as variation between source regions, land vs ocean, and the seasonality in the two hemispheres, there are some local differences mainly due to different observation times and locations. To provide context for comparison differences we examined CO volume mixing ratio variabilities in MOPITT data. For

example, within 500km area and 24 hours of a typical day in September, CO variations are about 12% and 15% in the lower and upper troposphere in North America, respectively.

The CO vertical profile retrievals from both MOPITT and TROPESS-CrIS are based on optimal estimation theory (Rodgers, 2000). The estimated radiance noise levels of the satellite instruments are propagated to retrieval measurement error (or precision). The a priori knowledge about the horizontal and vertical distributions from a CO climatology are also important

constraints in the optimal estimate process. The a priori CO profiles are used as initial guess profiles for both TROPESS and MOPITT CO retrievals. Different data processing teams use different a priori data. This will therefore cause differences in their retrieved profiles and the accompanying characteristic data. The steps for comparing TES-MOPITT CO profiles, i.e., how these are adjusted for the different a priori data, have been presented in Luo et al. (2007b). We selected a few days of TROPESS and MOPITT data over the four seasons of 2016, made inter-satellite CO retrieval comparisons following the steps

referenced above. The statistical comparison conclusions (details not shown) are similar for the selected days. They are also similar to a specific example case we show below.

Figure 1 shows coincident pairs of MOPITT and TROPESS-CrIS CO volume mixing ratios (VMRs) at the pressure level of 681 hPa for September 12, 2020. The lower troposphere 681 hPa is one of the forward model pressure levels used in TROPESS-CrIS retrievals defined via 12 levels between 1000 and 100 hPa uniformly in log(pressure). We apply coincidence criteria so

that the retrievals are within 24 hours and 500km of each other. TROPESS-CrIS CO concentrations are higher over eastern China and the biomass burning region of southern Africa and lower over the western US, but otherwise there is general agreement in global CO distribution patterns.

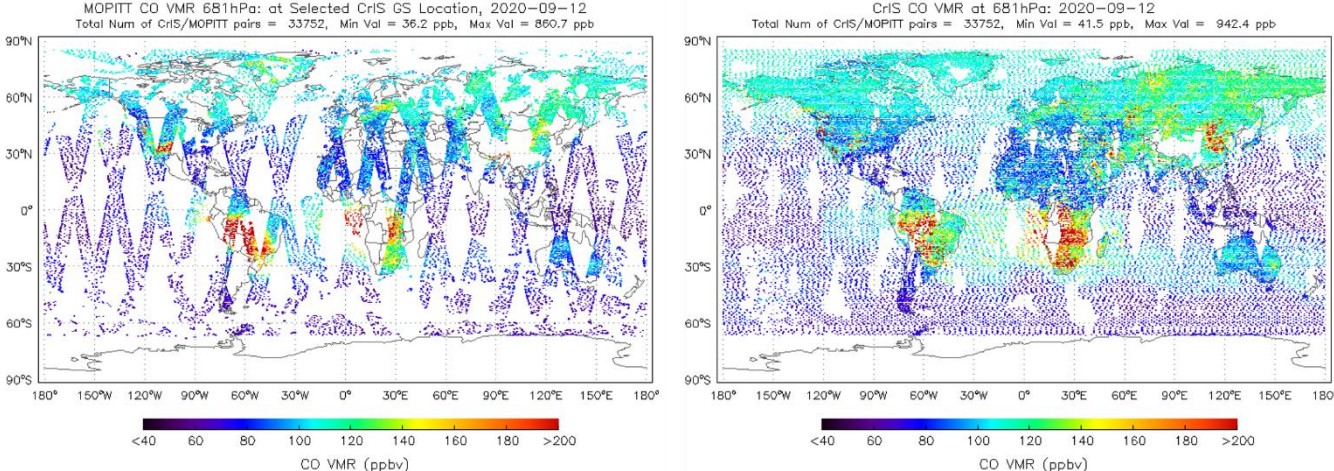

**Figure 1. MOPITT and TROPESS-CrIS CO profiles are matched in location (within 500 km) and time (24 hours). MOPITT CO**
**profiles are mapped to TROPESS-CrIS standard pressure levels. The CO Volume Mixing Ratio (VMR) in ppbv corresponding to**
**681 hPa are shown for Sept 12, 2020. The footprints are enlarged for illustration.**

MOPITT processing (V7 and later) uses a climatology from CAM-Chem (Lamarque et al, 2012) for the spatial and seasonal (but not interannual) variability in the CO a priori profiles. This is somewhat different from the MOZART climatology (Brasseur et al., 1998) used in the TROPESS algorithm. Both MOPITT and TROPESS-CrIS use the same vertical constraint (a priori covariance) of 30% uncertainty for CO parameters at all levels and a correlation length of 100 hPa between them in the troposphere. The use of the same prior covariance simplifies the intercomparison of satellite products (George et al., 2015).

Figure 2 shows the CO a priori VMRs ($x_a$) at 681 hPa for Sept 12, 2020 for the two instrument retrievals. From the MOZART model field, the TROPESS algorithm derives the monthly means over 10° latitude x 60° longitude blocks to extract CO profiles for the retrieval initial guess a priori profiles for CrIS; MOPITT interpolates the CAM-Chem model field (with 1.9° latitude x 2.5° longitude) at the observation location and time. The different climatology sources and how they are applied spatially results in many differences in $x_a$ when comparing coincident pairs. For example, as we show later, the $x_a$ used in MOPITT and TROPESS-CrIS retrievals are very different in magnitude for this day in the western US when several large wild fires occurred.

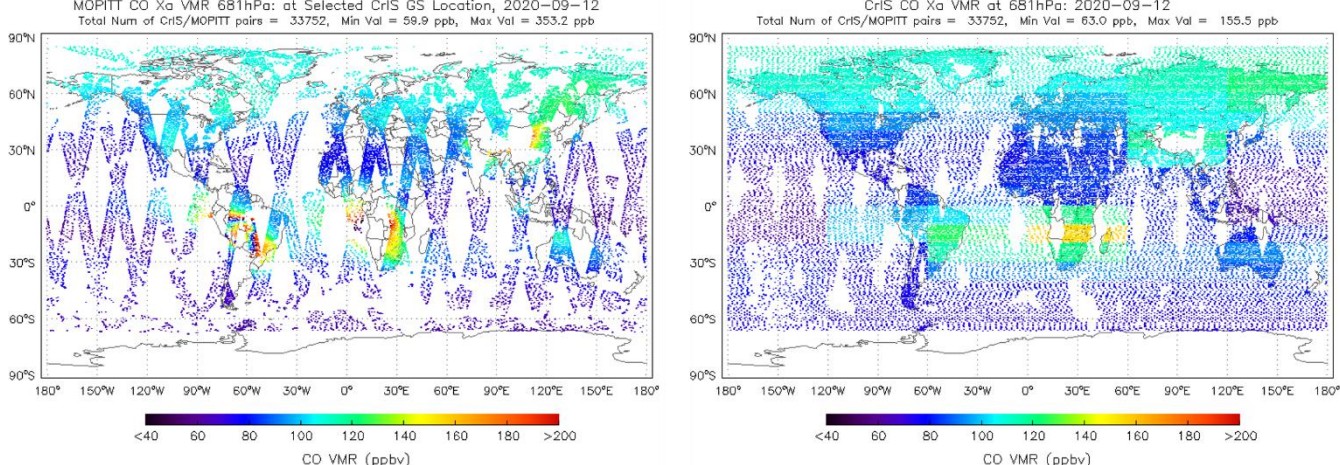

**Figure 2. The a priori CO profile VMRs at 681 hPa used in MOPITT and TROPESS-CrIS CO retrievals Sept 12, 2020, respectively. See text for climatology descriptions. MOPITT data use interpolation to observation local time and location while TROPESS-CrIS applies monthly means over latitude/longitude blocks to determine the a priori profiles.**

As described and illustrated in Worden et al.,2007 and Luo et al., 2007b, the trace gas profile retrievals are strongly influenced by the a priori data, especially for the nadir-viewing satellite instruments. For CO profile retrievals the degrees of freedom for signal (DOFS) are generally less than 2. This means the profiles only have a couple of independent information vertically. The following equation describes the relationship between a retrieved CO profile ($x_{\text{retv}}$) and the unknown "true profile" ($x_{\text{true}}$) assuming the known initial climatology state ($x_a$) is close to its "truth" with the uncertainty constraint (the linearization is valid):

$$x_{\text{retv}} = A\, x_{\text{true}} + (\text{I-A})\, x_a + e \qquad\qquad\qquad (1)$$

where A is the averaging kernel matrix describing the sensitivity of the retrieved state to the true state, and $e$ is the error mainly due to instrument measurement noise. The averaging kernel is determined by the sensitivity of the measurement to the retrieved CO state (Jacobian matrix) and the prior covariance matrix used to constrain the retrieved profile with only a couple of vertical degrees of freedom for signal (DOFS). The detailed linear retrieval estimate equation and the definition equations for A can be found in Rodgers (2000). We examine the DOFS of the two instrument CO retrievals for September 2020 (Figure 1). The DOFS for TROPESS-CrIS CO is larger than that of MOPITT CO by 0.1-0.2 (their global averaged DOFS are 1-2), indicating a slightly higher vertical resolution in TROPESS-CrIS CO retrieved profiles.

Since TROPESS-CrIS retrievals use the a priori CO profiles derived from a similar atmospheric model and the same constraints as MOPITT products, we can directly compare the retrieval products of the two data sets with relatively good agreement. For example, the global total CO column comparisons between TROPESS-CrIS and MOPITT shown in Figure 3 agree very well

in the zonal mean. However, the vertical integration of the CO profile to obtain total columns can average out potential disagreements in vertical sub-layers.

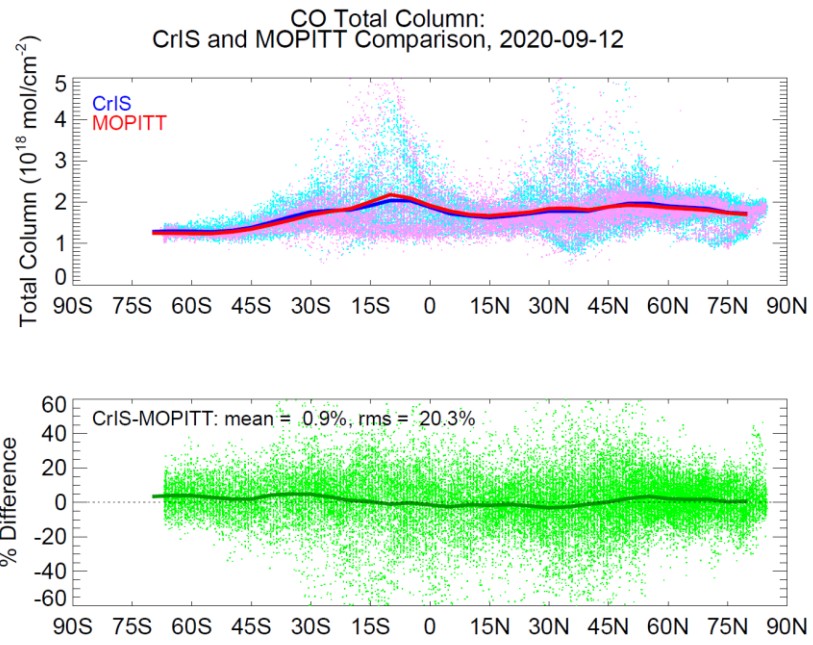

**Figure 3. Comparisons of CrIS and MOPITT CO total column retrievals, Sept 12, 2020.**

We present comparisons of TROPESS-CrIS and MOPITT CO profiles at different pressure levels. We follow the procedure and equations described in Luo et al., 2007b for making direct comparisons, by adjusting the MOPITT a priori profile $x_a$ to that of CrIS and smoothing the CrIS profiles with the MOPITT averaging kernel (A). The smoothing procedure is applied to TROPESS-CrIS CO profiles because their DOFS, e.g., zonally averaged, are slightly larger than that of MOPITT. As we emphasized, the influences of the different a priori data used in the profile retrievals will contribute to the disagreement of the

trace gas profile products provided by different retrieval teams. Examples of the results are shown in Figure 4 at 681 hPa and 215 hPa. Table 2 summarizes the comparison statistics for Sept 12, 2020.

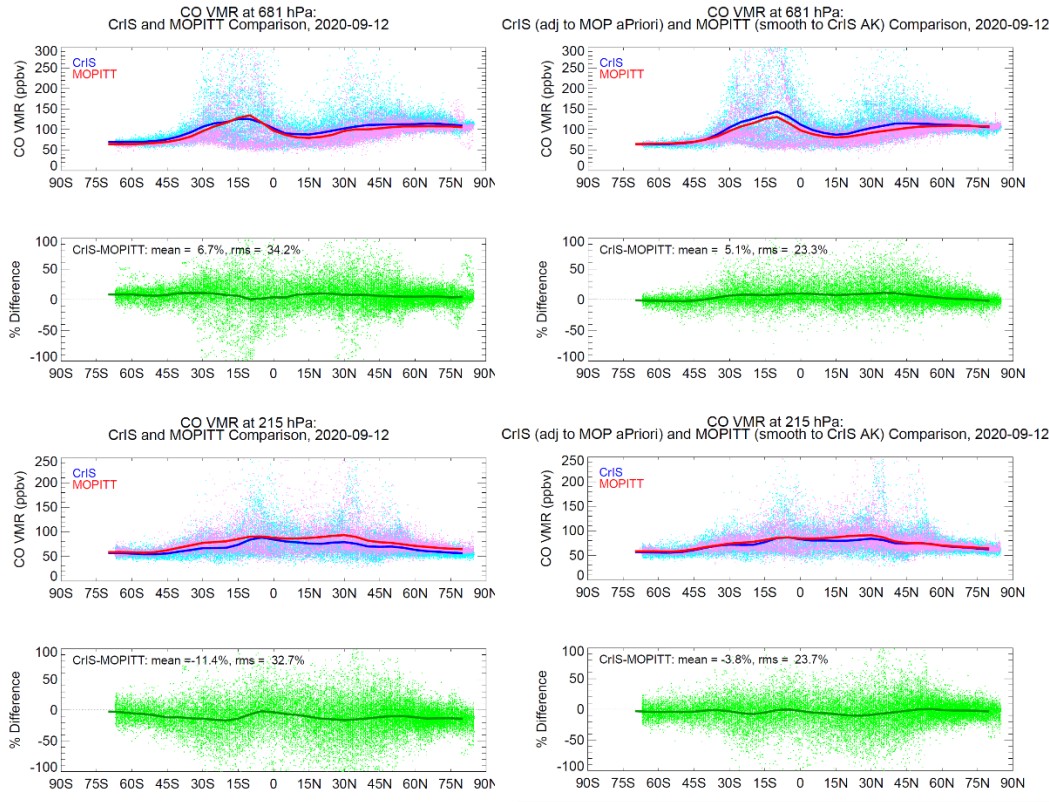

**Figure 4. TROPESS-CrIS and MOPITT CO comparisons at 681 hPa (top) and 215 hPa (bottom), Sept 12, 2020. The left column shows the direct comparisons, and the right column show the comparison of CrIS adjusted to MOPITT a priori profile Xa and MOPITT smoothed by CrIS Averaging kernel.**

**Table 2.** TROPESS-CrIS and MOPITT CO comparison summary, Sept 12, 2020.

| | | A Priori $x_a$ | Direct Comp | Adj $x_a$ | Adj $x_a$ & AK | Retv Err % | | Retv Precision % | |
|---|---|---|---|---|---|---|---|---|---|
| | | | | | | CrIS | MOP | CrIS | MOP |
| Total Column | % Diff | | 0.9% | | | | | | |
| | % RMS | | 20% | | | 10% | 8.6% | | |
| 681 hPa | % Diff | -1% | 6.7% | 6% | 5.1% | | | | |
| | % RMS | 23% | 35% | 27% | 23% | 25% | 12% | 4.6% | 2% |
| 215 hPa | % Diff | -10.8% | -11.4% | -4.7% | -3.8% | | | | |
| | % RMS | 9.7% | 33% | 31% | 23% | 25% | 12% | 6% | 3% |

Note: Diff = TROPESS-CrIS - MOPITT CO; RMS=root mean square of the Diff.

Each step in the comparison process reduced disagreement between TROPESS-CrIS and MOPITT CO, as expected. At 681 hPa, their direct globally averaged difference was 6.7% (CrIS minus MOPITT) with 35% RMS; this difference was reduced to 6% with 27% RMS with the a priori adjustment, and further reduced to 5.1% with 23% RMS with the application of the averaging kernel. At 215 hPa, the three step comparisons are -11% (33% RMS), -4.7% (31% RMS), and -3.8% (23% RMS). We also listed the percent retrieval error (Retv Err %) and precision (Retv Precision %). The above quoted mean differences are comparable to the retrieval precisions and within the CO natural variability of 12-15% mentioned above.

We note that even after adjusted two data sets for the slight differences in the a priori assumptions, compared to MOPITT, TROPESS-CrIS CO VMRs are still ~5% higher in the lower troposphere and ~4% lower in the upper troposphere. This result is in good agreement with previous work comparing satellite CO profiles to in-situ observations (Luo et al., 2007a, Hegarty et al., 2021, Deeter et al, 2022, Worden et al, 2022).

## 3 Comparisons of model CO simulations to the satellite CO retrievals

The above analyses of the two satellite CO profile retrieval comparisons have shown that satellite data users should not treat retrieved data products as the "truth". The retrieval characteristic data, e.g., the a priori profiles and the averaging kernels derived from the retrieval processes are key parameters in the applications. Here we briefly describe the GISS Earth System model (ModelE2) as an example in this study. In the next two sections, we illustrate the proper use of the retrieval data sets in model evaluations.

### 3.1 GISS Earth System model

The GISS ModelE2 simulates the interactions between the different components of the Earth system. The model can be used to study a wide range of climate phenomena, including the impacts of greenhouse gases, aerosols, and other atmospheric pollutants on the climate. We used the NASA GISS-E2 version described in Kelley et al. (2020), with prescribed sea-surface temperatures and interactive chemistry. Aerosols are coupled to the tropospheric chemistry scheme which includes inorganic chemistry of $O_x$, $NO_x$, $HO_x$, CO, and organic chemistry of $CH_4$ and higher hydrocarbons.

Anthropogenic fluxes come from the Community Emissions Data System inventory (Hoesly et al., 2018) and sea salt, dimethyl sulfide, isoprene and dust emission fluxes are calculated interactively. All other forcings, such as solar, volcanic (prescribed as stratospheric AOD and aerosol size) and land-use follow the CMIP6 protocol (Eyring et al., 2016). Biomass burning emissions and injection heights are prescribed from the Global Fire Assimilation System (GFAS) (Kaiser et al., 2012; Remy et al., 2017) at a daily time step, rather than monthly averages and boundary layer distribution of fire emissions used in base CMIP6 configuration. Emission sources in September 2020 mainly due to the intense wildfires in Western US and the

background biomass emissions. These are shown for CO in Figure 5 for Sept 12, 2020, for both the original GFAS 0.1º x 0.1º

resolution and the ModelE2 2.0º x 2.5º resolution, obtained by re-binning from the higher resolution GFAS grid to the coarse ModelE2 grid while conserving total emissions. The CO emission hot spots are associated with the reported wildfires in the news, such as the Bobcat fire in Angeles National Forest and the Creek Fire in the Sierra National Forest (https://en.wikipedia.org/wiki/2020_California_wildfires). Several large wildfires also occurred in the States of Oregon and Washington (https://en.wikipedia.org/wiki/2020_Western_United_States_wildfire_season).

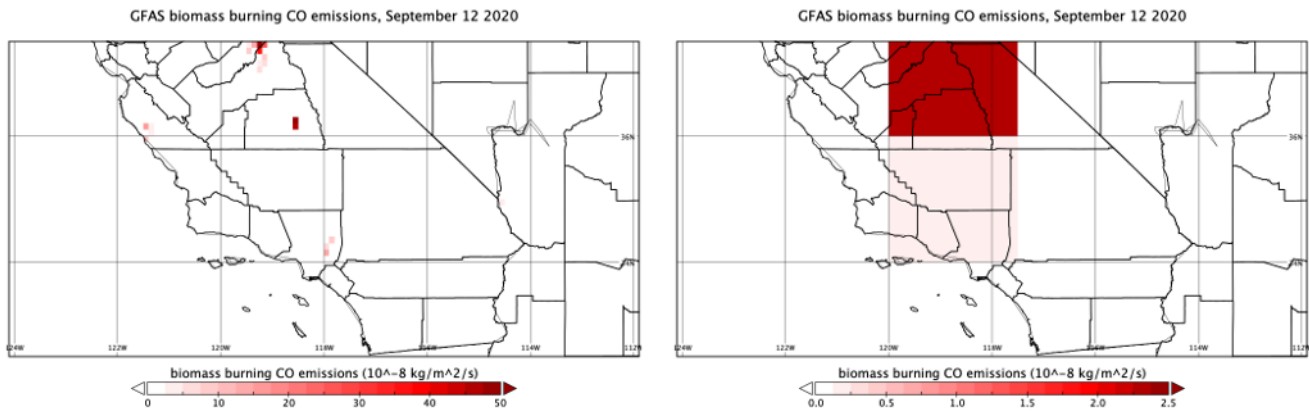

**Figure 5. CO emission sources on Sept 12, 2020. Left, The wildfire flux of CO in the Global Fire Assimilation System (GFAS) source files at 0.1 x 0.1 degree latitude/longitude. Right, the same GFAS CO emissions converted to ModelE input at 2 x 2.5 degree latitude/longitude, noting the different scale.**

Time-evolution of the distributions of enhanced CO due to fires depends on emission fluxes and the transport processes in the atmosphere. We nudged GISS ModelE2 horizontal winds to National Centers for Environmental Prediction (NCEP) reanalysis (Kalnay et al., 1996), driving the trace gas transport away from the fire source areas. Some model parameters that determine the gas initial locations, such as the injection heights over fires and aerosol scheme are subjects of model parameter evaluations using in-situ and remote observations. We leave these detailed ModelE2 investigations to another publication by Field et al.

(submitted). The CO model output used in this paper are at 2 º x 2.5 º latitude by longitude and hourly intervals. They were sampled at the geolocations and times of satellite profile retrievals for comparisons.

### 3.2 Proper comparisons of model to the satellite CO retrievals

Here we use CO data during the Western US wildfires inSeptember 2020 to illustrate the steps of comparing GISS ModelE2 CO simulations to CrIS and MOPITT CO observations. This follows other model-satellite comparison with biomass burning

as a key CO source for a single wildfire event (Field et al., 2016), and at seasonal (Liu et al., 2010; Field et al., 2015) and interannual scales (Strode et al., 2016), for example.

As we described in section 2, TROPESS-CrIS and MOPITT provide CO profile retrievals with DOFS of 1-2 over clear sky conditions. Figure 6 shows CO VMR distributions from TROPESS-CrIS and MOPITT observations taken on September 12, 2020 over US in the middle troposphere at about 450 hPa. This vertical range is the sensitivity peak of the satellite nadir

observations to the CO local concentrations. TROPESS-CrIS and MOPITT CO maps show very good agreement in highlighting the huge CO plumes originated from the catastrophic wildfires (e.g., https://www.nasa.gov/feature/jpl/nasa-monitors-carbon-monoxide-from-california-wildfires). In Figure 7, we also show the satellite CO maps near the surface at 750 hPa where the outstanding high CO VMRs are most likely closer to the emission sources – the burning area at the ground over land. Over the Ocean, the high CO are mostly due to the combination of tracer transport from its origin and the effect of vertical

smoothing in retrievals.

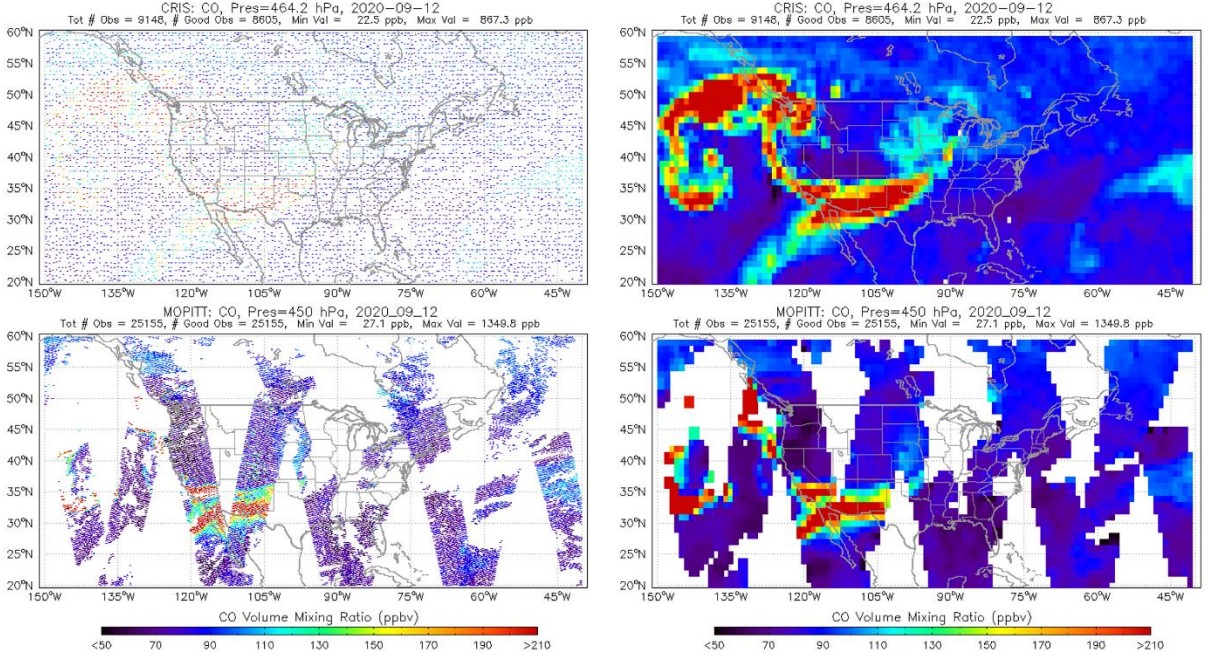

**Figure 6. The CO value colored dots (left) at observations locations, and 1x1 degree latitude/longitude averaged (right) CO VMR maps at 464.2 hPa for TROPESS-CrIS (top) and 450 hPa for MOPITT (bottom).**

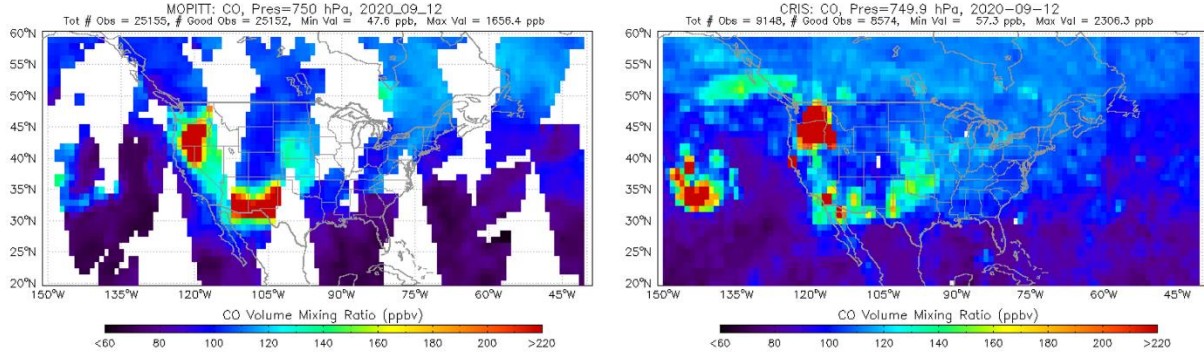

**Figure 7. The 1x1 degree latitude/longitude averaged CO VMR maps at 750 hPa for MOPITT (left) and 749.9 hPa for TROPESS-CrIS (right).**

There are some noticeable differences in CO distributions at two pressure levels (Figures 6 and 7). For example, the high CO over the Pacific Ocean in TROPESS-CrIS maps is less apparent in MOPITT distributions. In additions to the maximum of 24
230   hours' time and the exact footprint differences (comparing the dot maps in Figure 1 over W US), the instrument noise contributing to the retrieval errors is a factor too. The precision (measurement error due to noise in spectral radiances) and the total retrieval error for TROPESS-CrIS (Bowman 2006) are over twice of that for MOPITT (Table 2). One of the reasons is perhaps that the MOPITT retrievals are flagged as missing due to the thick smoke (Deeter, 2022).

The ModelE2 CO field at 2x2.5 lat/lon grid and one-hour time interval described in section 3 are sampled at the satellite
observation location and times. The next step is to calculate the "retrieved profile" assuming the model profile is the "truth" following equation (1). This "retrieved profile" obtained via applying retrieval operator is the proper way of comparing the model to the satellite data retrievals.

In ModelE2-CrIS CO comparisons, the left panels of Figure 8 show the model "raw" CO maps with model time/location sampling at SNPP/CrIS observations. In the right panels of Figue 8, the model "retrieved" CO maps, described above for Sept
12, 2020 at pressures 464.2 hPa and 749.9 hPa respectively are shown. At the pressure level near the surface (749 hPa or about 2.5 km), the CO emission source distributions and the near surface transport effects are seen in the model simulations. The model "raw" CO distributions exhibit strong CO emissions from multiple wildfire sources in the western US States. It also demonstrated fire plume transport patterns similar to the CO maps of TROPESS-CrIS and MOPITT, e.g., a spiraling segment to the Pacific Ocean and a separate eastward segment (Figure 7). It appears that the model meteorological winds near surface
effectively transport the fire generated pollutants over long distances. These model features are mostly confined to 749.9 hPa, with an isolated enhancement at 464.2 hPa of up to ~170 ppbv only over the US Midwest. The proper model-satellite CO concentration comparison is to compare the "retrieved" model CO (right panels in Figure 8) with CrIS CO at the same pressure levels (Figures 6 and 7). At 464.2 hPa, the model CO feature is still apparent after applying the TROPESS-CrIS retrieval operator, but less pronounced than the raw CO, peaking at ~150 ppbv. Closer to the surface at 749.9 hPa, the effect of the

retrieval operator is greater, with most of the enhanced model CO absent except for an isolated feature over the Pacific Northwest. These comparisons illustrate the relatively low-sensitivities in satellite profile retrievals especially near the surface, and their effect on the 'raw' model profiles. The smoke injection heights prescribed from GFAS were also likely to be underestimated given the intensity of the fires leading up to September 12, 2020 (Lassman et al., 2023).

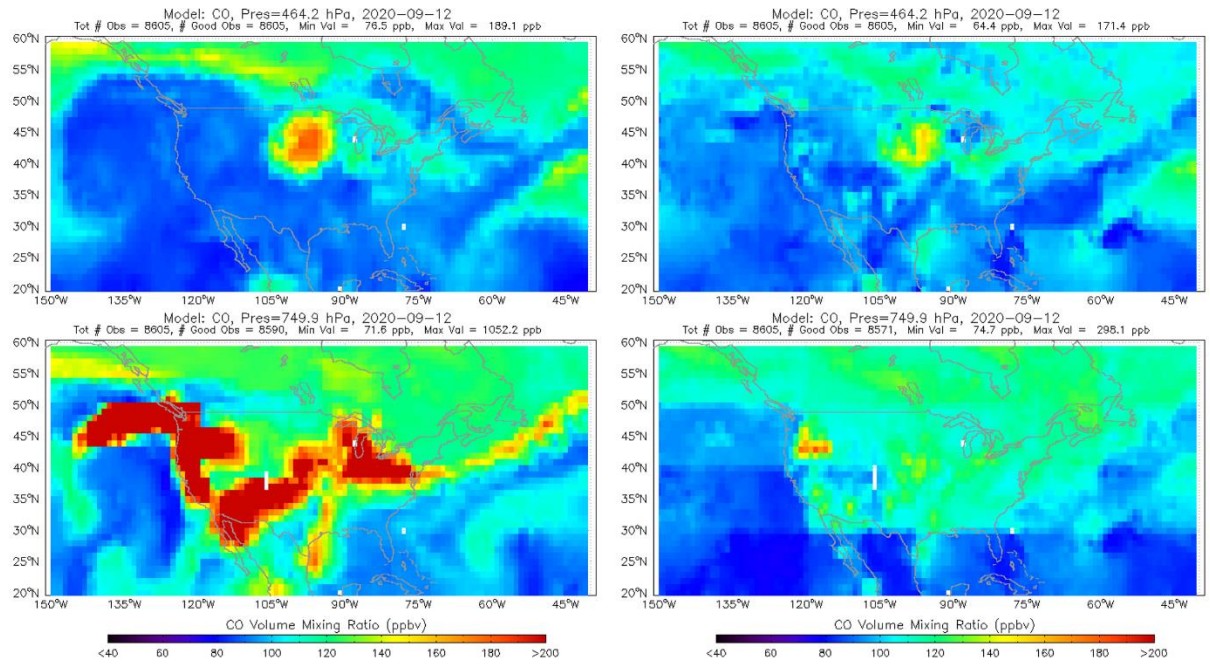

**Figure 8. GISS model CO VMRs for Sept 12, 2020 at 464 hPa (top) and 749.9 hPa (bottom). Model CO profiles are sampled at SNPP/CrIS time and footprints and averaged at 1x1 degree lat/lon grids (left). The right panels show the "retrieved" model CO profiles from their "raw" data using TROPESS-CrIS retrieval operator (equation 1).**

Similarly, the model to MOPITT CO comparisons shown in Figure 9 show almost the same conclusions as model to
260 TROPESS-CrIS CO comparisons. In addition to the CO distribution patterns at two pressure levels that we discussed above, the differences in model "raw" map and the model "retrieved" map after using the satellite retrieval operator is obvious, especially in lower troposphere (749.9 hPa), although there the model CO enhancement remains more apparent compared to the TROPESS-CrIS to retrieved model CO possibly because of MOPITT's slightly greater retrieval sensitivity near the surface. As we reference the satellite data a priori CO VMRs shown in Figure 2, we know in the lower troposphere, the influence of
265 the a priori data to the retrieved profiles is very large (the second term of equation 1). In the next section, we use the averaging kernels of an example profile to demonstrate this influence.

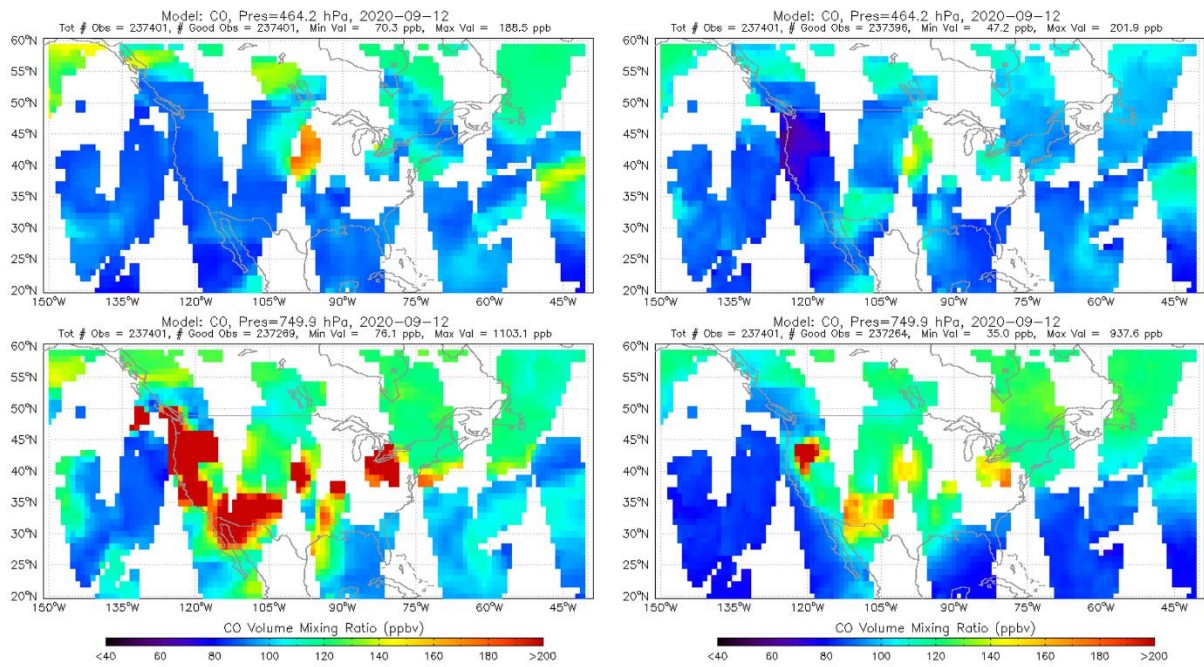

**Figure 9. Similar to Figure 8, the GISS model-MOPITT sampling and retrieval operator application.**

**4 Discussion of model-satellite profile comparisons**

Equation (1) in section 2 presents a simple relationship between the true species profile and the retrieved one. It assumes that the initial guess of the profile in the iterative optimal retrieval process is close to the climatology mean (the a priori $x_a$) described by the a priori constraint matrix defining the variability of the mean. For a given spectral radiance satellite measurement, a different a priori profile could result in different retrieved profile even if using the same constraint matrix. Note in TROPESS-CrIS and the MOPITT CO retrievals, the a priori (also used as the initial guess profiles) are chosen differently for the two

project teams respectively (Figure 2). We use one fire scenario to discuss the details.

From the CO fire maps in Figures 6-9, Sept 12, 2020, we selected one CO enhancement in Southern California that is common to the CrIS and MOPITT data. This is likely influenced by emissions from the Bobcat fire near Mt Wilson observatory burning in the foothill area in multi cities (https://earthobservatory.nasa.gov/images/147324/bobcat-fire-scorches-southern-california). Using the fire center at latitude and longitude of 34.2N, 118W degrees, we identified one profile from TROPESS-CrIS and

MOPITT observations respectively. We use the criteria of the observation location that was among closest to the fire and had the maximum DOFS in CO retrievals. Due mainly to the remaining mismatch in location and time near the fires, TROPESS-CrIS CO values are ~2X the MOPITT CO in the mid-troposphere.

Figure 10 shows the selected CrIS and the matched model CO profile comparison. Figure 11 shows the selected MOPITT and its matched model CO profile comparison. In these comparison cases, both TROPESS-CrIS and MOPITT profiles show very

high CO in the mid-troposphere (700-300 hPa), while neither the original or the 'retrieved' model profiles display any CO enhancement at these higher altitudes. Since the satellite AK peaks are in these mid-troposphere levels, the model "retrievals" are only moderately increased compared to the original one, indicating the very weak CO plume transports vertically or a weaker CO emission in model setup near the surface.

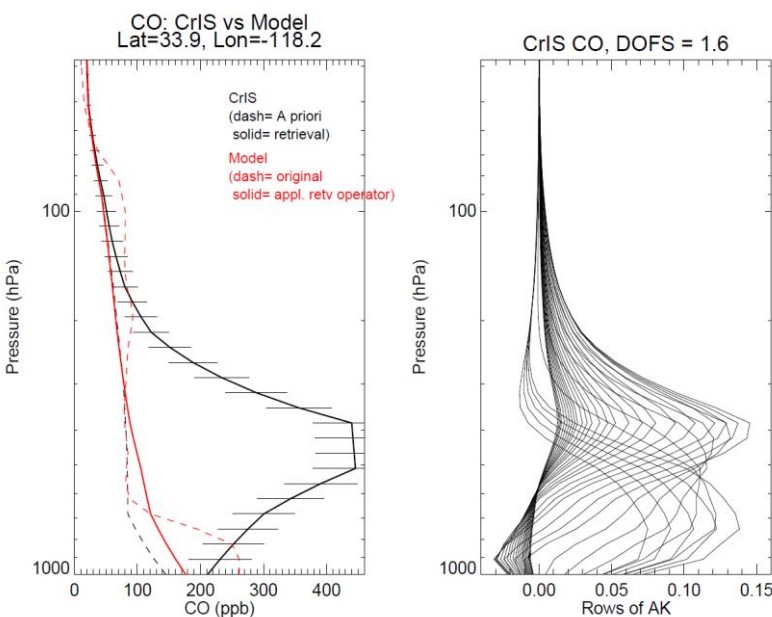

Figure 10. CO profile comparisons at 34.2N, 118W on September 12, 2020 near the Bobcat fire center. The left panel shows (1) TROPESS-CrIS CO retrieved with error bars and the a priori (dash) profiles in black, and (2) the matched original model CO profile (dash red) and the "model retrieved" profile after applying CrIS retrieval operator (solid red). The right panel shows the CrIS averaging kernels.

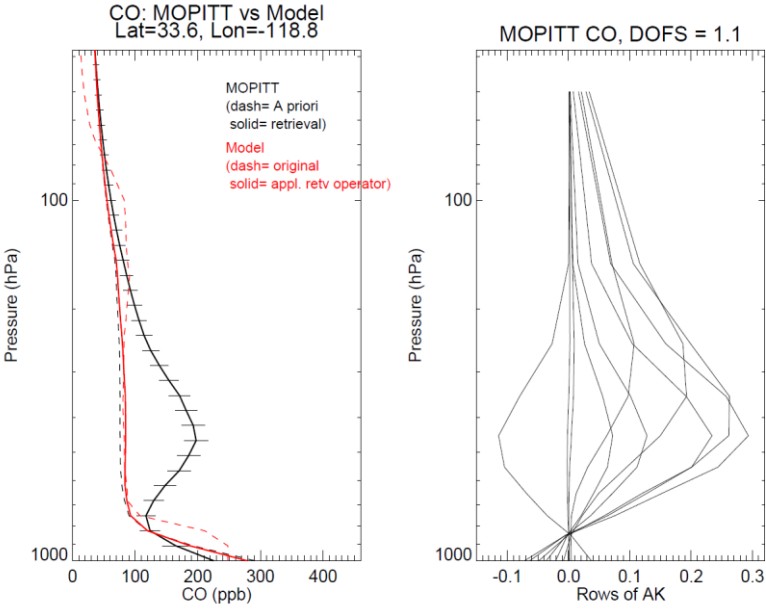

Figure 11. Similar to Fig 10. The satellite CO profile is from MOPITT.

Near the surface, according to the averaging kernels of the two instruments, the satellite retrieved CO profiles should be insensitive to the CO emissions. The retrieved profile themselves were therefore pulled over to the values of the a priori. We also noted that the a priori guess of TROPESS-CrIS and MOPITT CO were different due to the different ways that the two teams used to derive them (section 2) – TROPESS-CrIS and MOPITT CO surface a priori values are less and greater than 200 ppb respectively for the case discussed here. The "retrieved" model CO near surface are therefore pulled to the a priori respectively (the solid red in Figures 10 and 11). These changes in the model CO maps are also seen in the right panels of Figures 8 and 9.

We also examined total column comparisons between the corresponding model and satellite profiles in Figures 10 and 11. Model CO total column ($2.4 \times 10^{18}$ mol/cm$^2$) underestimated column COs retrieved by the TROPESS-CrIS ($5 \times 10^{18}$ mol/cm$^2$) and MOPITT ($3 \times 10^{18}$ mol/cm$^2$).

Figure 12 shows the result of using the MOPITT CO a priori profile near the Bobcat fire as the initial guess and a priori (black dash line, Figure 11) in the collocated CrIS CO profile. Compared to the CrIS retrieval using the TROPESS $x_a$ (Figure 10), the retrieval using the MOPITT $x_a$ resulted in a different CO profile, especially near the surface. This is due to dominant contributions from the a priori near the surface where the averaging kernels has lower sensitivity to the true profile. Based on the results of Kulawik et al., (2008), we do not expect full retrievals of profiles that assume an a priori to match exactly to retrievals where that same prior is "swapped" in a single step following the retrieval iterations (the method described in Section 2) with a different a priori due to non-linearities in the retrieval process.

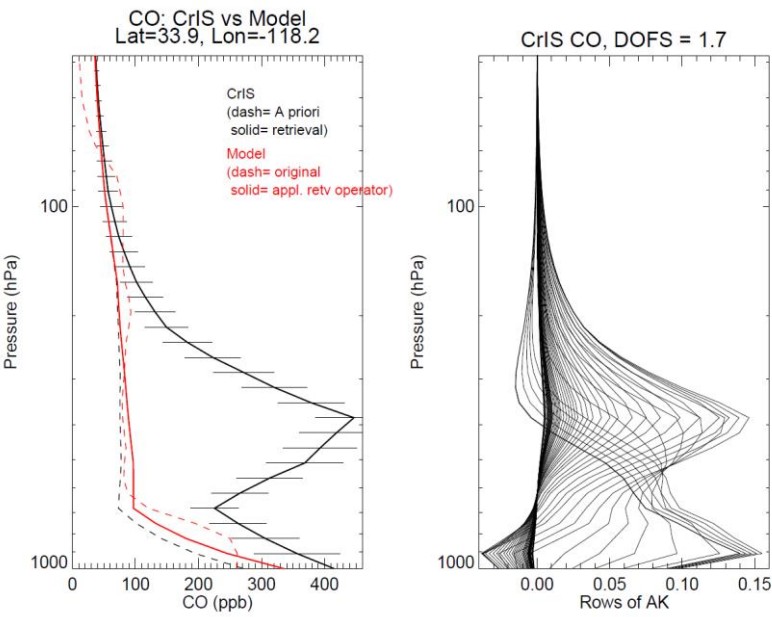

**Figure 12. In left panel, the CrIS CO retrieved profile is generated from MOPITT a priori, also as the initial guess (black); the model "retrieved" profile (solid red) is derived using this new TROPESS-CrIS retrieval operator. Right panel shows the corresponding TROPESS-CrIS CO averaging kernel for this new retrieval.**

Figure 13 shows comparisons of CrIS CO profiles generated via three retrieval configurations. In the left panel, two dashed lines show the different a priori/initial guess profiles from the TROPESS and MOPITT algorithms. The corresponding CrIS CO retrievals are shown in solid black and red profiles. At 700-300 hPa, compared to the two very similar a priori profiles, we see strongly enhanced CO layers in both retrievals, indicating the dominant observable signal from  fireenhanced CO in the mid/lower-troposphere. Near the surface we see the dominant effect of the a priori in the retrieved CO values.

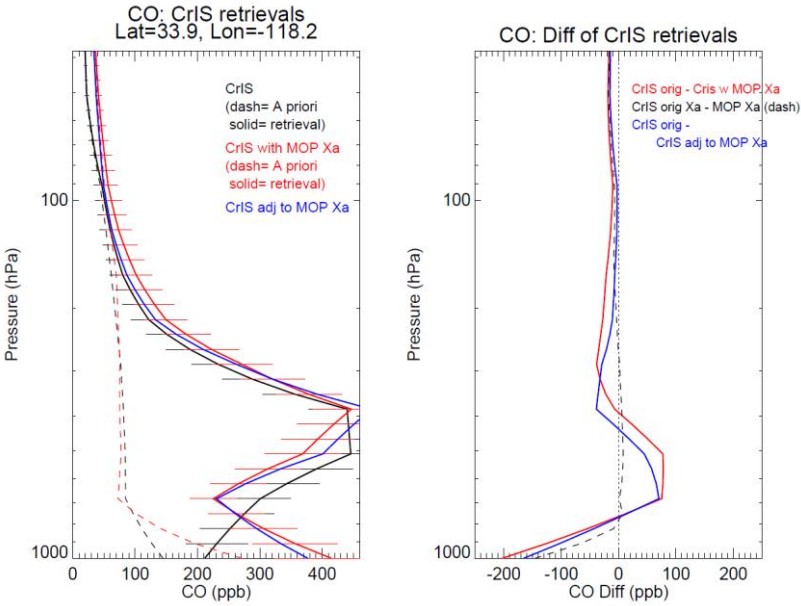

**Figure 13. Left panel shows the overlaid CrIS CO retrieved and the a priori profile (black), CrIS CO retrieved using MOPITT a priori profile (red),  and the CrIS CO adjusted with MOPITT a priori profile using Equation (1) (blue).  Right panel shows their comparisons, CrIS CO retrieval minus CrIS CO retrieval using MOPITT a priori (red), CrIS a priori Xa minus MOPITT Xa (dashed), and CrIS CO retrieval minus CrIS CO adjusted to MOPITT X (blue).**

The third way to derive a retrieved CO profile is via swapping the a priori $x_a$.  The blue profile in Figure 13 left panel is obtained by simply adjusting the TROPESS-CrIS CO a priori from the original retrieval to that of MOPITT (Luo et al., 2007b). This profile (blue) is very similar to the full retrieved CO profile (red), however, differences remain as shown in the right panel of Figure 13. As found in Kulawik et al. (2008), the profile differences from these two approaches are small and are due to retrieval non-linearities.  These profile differences can be evaluated using the retrieval precision due to instrument noise terms (a few percent listed in Table 2), which demonstrates the validity of the simpler approach (i.e., swapping the a priori) before comparing profiles from different instruments or retrievals.

### 5 Summary

The TROPESS algorithm including the a priori assumptions inherited from the TES project has been used to retrieve several atmospheric species profiles from CrIS and other satellite nadir spectral measurements. Here we made the comparisons of TROPESS-CrIS and MOPITT CO retrieved profiles globally, in steps of adjusting their a priori and vertical smoothing effects. A better agreement between the two satellite data sets is achieved at the last step. The slight biases of TROPESS-CrIS CO compared to MOPITT are about 5% in the lower troposphere and -3% in the upper troposphere. The RMS of the above bias is 23% which can mostly be explained by the CO 12-15% variabilities in 24 hours and 500 km area, and the measurement errors of 2-6% of the two instruments due to their radiance measurement noises.

Using the GISS ModelE2, we illustrated the proper method for making model-satellite CO profile retrieval comparisons, a necessary step in evaluating model-crucial parameters. For data taken during the historical large wildfires in W US, September 2020, the retrieval a priori dominates near the surface where the satellite measurements have less sensitivity causing the model "retrieved" CO to move toward the a priori; in the mid-troposphere where TROPESS-CrIS and MOPITT show the maximum sensitivity to the true concentrations in their retrievals, the model "retrieval" departs from the satellite retrievals. This disagreement indicates unmatched CO emission locations/times and (or) yet to be improved tracer transport schemes in GISS model, particularly in the vertical. We use the CO vertical profiles near the Bobcat fire center to examine this model-satellite comparison situation.

Finally, the TROPESS-CrIS and MOPITT single CO profile retrievals are used to illustrate the comparison of adjusting to a common a priori for the retrievals mathematically vs. carrying out the retrievals end-to-end. We found the swapping the a priori mathematically works well.

### Data availability

TROPESS-CrIS CO products are available via the GES DISC from the NASA TRopospheric Ozone and its Precursors from Earth System Sounding (TROPESS) project at https://doi.org/10.5067/I1NONOEPXLHS (Bowman, 2021). The MOPITT Version 9 products are available from NASA through the Earthdata portal https://asdc.larc.nasa.gov/project/MOPITT/MOP02T_9.

### Author contributions

ML planed and carried the study. ML and HMW discussed and wrote the manuscripts for the TROPESS and MOPITT CO comparisons. RDF and KT provided GISS ModelE2 results, description and analyses. RDF, KT and GSE contributed discussions on model-satellite comparisons. RF edited the manuscript. All authors reviewed manuscript.

### Competing interests

At least one of the (co-)authors is a member of the editorial board of Atmospheric Measurement Techniques.

**Disclaimer**

Publisher's note: Copernicus Publications remains neutral with regard to jurisdictional claims in published maps and institutional affiliations.

**Acknowledgements**

We would like to thank the TROPESS science and software teams for their algorithm insights, supportive discussions and data processing. We especially thank the discussions with Susan Kulawik on retrieval algorithms and the setup of the MUSES processing codes by Valentin Kantchev.

**Financial support**

Part of this research was carried out at the Jet Propulsion Laboratory, California Institute of Technology, under contract with the National Aeronautics and Space Administration via the TRopospheric Ozone and its Precursors from Earth System Sounding (TROPESS) project. The project is also supported by NASA grant (80NSSC18K0166).

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
