# Peer review of "TROPESS-CrIS CO single pixel vertical profiles: Intercomparisons with MOPITT and model comparisons for 2020 US Western wildfires"

_EGUsphere, 2023_

## Author Response (AR1)

**RC1**: 'Comment on egusphere-2023-1369', Anonymous Referee #2, 20 Oct 2023

This manuscript compares CrIS-CO, MOPPIT-CO and model CO simulations for wildfire emissions in the western US. It is a valuable documentation on how CO retrievals from different instruments should be compared and evaluated against model simulations. From the comparison with simulations from GISS ModelE2, the authors suggest the model may have underestimated the smoke injection height. The topic fits well into the scope of AMT. Some major and minor revisions will be necessary before re-evaluation and publication.

Thanks for the detailed review of our manuscript by Referee #2.

1. Presentation related: (1) currently the methods, datasets, and results are mixed together in Section 2, 3, and 4, I would suggest to add a "data and methods" section to specifically introducing used datasets and model simulations; (2) The numbers in Figure 6, 7, 8, and 9 are hard to read due to low quality graphics. please consider using higher solution images;

(1) Sections 2, 3, and 4 presented two topics, TROPESS-CrIS and MOPITT CO comparisons (section 2) and using satellite data (CrIS and MOPITT CO respectively) to evaluate GISS model simulations of CO (section 3 and 4).  We accepted the reviewer's suggestion, and changed the presentations into to two sections to present the above two topics and used sub-sections for data description, method and comparisons etc.

(2) Figures 6-9 are replaced by the higher quality images.

2. The satellite data are compared using zonal mean, while a point-by-point comparison is not difficult to do and will be more helpful to understand the discrepancies. The spatial heterogeneity of CO distributions is large and zonal mean is therefore not a good way of making comparison. In addition, do you use nighttime retrievals. When you select data-pair observed less than 24 hours apart, will you be comparing day time to nighttime data?

All the comparison values in figures 3 & 4 (overlaid CO VMRs and the % diffs) are done for the matching pairs of the CrIS and MOPITT data globally. The "zonal mean" lines in the figure are simply used to illustrate the averaged effect in the two-dimensional plots. The comparison statistic in Table 2 (Table 1 in the original submitted version) are calculated using all comparison pairs globally. We think the use of CO percentage (%) differences are better than using volume mixing ratio directly because it considered regions of both enhanced and background COs.  Yes, we included nighttime data from the two instruments. The main criterion for selecting the CO pairs are the closest in distance within 24 hrs. We examine the histograms for the distances and the time diffs. Here are the results: (1) 45% and 77% comparison pairs are within 50km and 200km with peak distribution at 15.4km, and (2) 41% comparison pairs are within 6 hrs with peak distribution at 3-5 hrs.

3. I would suggest the authors to carry out one more experiment to compare with a different and may be better performed model simulations, such as the CAMS CO simulations. CAMS CO has assimilated IASI and MOPITT. Theoretically, the CrIS CO should close to CAMS simulations after applying the a priori and AK correction.

CAMS CO: https://ads.atmosphere.copernicus.eu/cdsapp#!/dataset/cams-global-atmospheric-composition-forecasts?tab=overview

Thanks for the suggestion of evaluating the satellite data - assimilated model analyses with the other satellite retrievals. And thanks for introducing the CAMS products. We agree that the assimilated products perhaps resemble the true atmosphere tracers more closely compared to a boundary emission driven transport model. A multi-model comparison was beyond the scope of this work, and is a good topic for a future study.

4. The vertical sensitivity information is not shown until Figure 10. Such information should be introduced earlier to justify the use of 681hPa when comparing MOPITT and CrIS in Figure 1, and the use of two different pressure levels in Figure 4.

Yes, we ignored the statement and perhaps a figure to show the vertical information in CO retrievals from the two observations. We added statement about the Degree of Freedom of Signals (DOFS) in CrIS and MOPITT CO retrievals and explained the selections of 681 and 215 hPa in our analyses. We referred related papers/documents for TROPESS-CrIS and MOPITT for averaging kernel plots in Section 2.

Specific comments:

L29-30, CO is currently measurable using 2.3um in the SWIR and 4.7um in the TIR. Please correct.

Thanks for catching this. We made the correction.

L30-32, this is not a full list. CO retrieval data from recent missions (e.g., GOSAT-2, FengYun-4B) need to be included. Please see:

GOSAT-2: https://amt.copernicus.org/articles/15/3401/2022/

FengYun-4B: https://amt.copernicus.org/articles/16/3059/2023/

Thanks for the info.  We added these CO data references.

L42: "for wildfire events September 2020" please rephrase.

Changed to "for September 2020 wildfire events in the Western US."

L53: "at a reduced global sampling– one every 0.8 degrees latitude and longitude box", what is the reason to reduce the sampling? Computational costs? How this may affect the comparison?

Yes, the decision is based on the limited processing resources. For daily global TROPESS/CrIS-MOPITT comparisons, we obtained over 20K data pairs – sufficient for statistical conclusions.

L54: "CriS" -> CrIS

Corrected.

L74: "For example, within 500km area and 24 hours", as mentioned above, will this lead to data-pairs of daytime and nighttime observations?

Both day and night MOPITT CO data are included in the calculation. Yes, data-pairs will have day-night pairs.  The closest in distance data pairs were selected within 24 hours.

L90: can you justify the use of 681 hPa here for comparison?

We added "The lower troposphere 681 hPa is one of the forward model pressure levels used in TROPESS-CrIS retrievals defined via 12 levels between 1000 and 100 hPa uniformly in log(pressure)."

L98: derives

Corrected.

L119: "TROPESS-CrIS retrievals use similar a priori CO profiles and constraints as MOPITT products". You mentioned that they use different model simulations and different sampling method. If this is true, the a priori can be very different.

Yes, the a priori profiles are different shown in Figure 2. The equation used to derive the a priori constraint is the same for the two retrievals. We presented step 1 to adjust the a priori profile of MOPITT to that of TROPESS-CrIS before making the comparison.

We edited the text to make the description clearer.

L127: "Luo et al., 2007b", please check all citation format through the paper.

Corrected and checked.

Figure 4: Please show averaging kernel information to justify the use of 681hPa and 215hPa for the comparison.

The averaging kernels are scene dependent. We added descriptions of degree of freedom for signal (DOFS) of the two instrument retrievals with a good fractions of scenes having DOFS>1. The two pressure levels (681 and 215 hPa) are the original forward model pressures of TROPESS, and they represent typical lower troposphere and upper troposphere respectively.

Figures 10 and 11: The vertical profiles of mixing ratio between satellite and model look very different. How are the total CO columns, are they close?

We calculated the total CO columns. Yes, the CO columns between satellite and model are very different – model under estimated CO. The values of the columns are ~5, 3, 2.4 x 10^18 (mol/cm2) for CrIS, MOPITT, and the model respectively.  We updated text with the descriptions of CO columns.

L282-284: too long. Please rephrase.

Original text: We did an experiment using the MOPITT CO a priori profile near the Bobcat fire as the initial guess and the a priori (Figure 11 black dash) to retrieve the CrIS CO profile also near the Bobcat fire. Figure 12 shows the result. Compared to the CrIS retrieval using the TROPESS $x_a$ (Figure 10), this new retrieval resulted in a different CO profile, especially near the surface where the averaging kernels shows lower sensitivity to the true profile, resulting in a dominant contribution from the a priori.

New text: Figure 12 shows the result of using the MOPITT CO a priori profile near the Bobcat fire as the initial guess and a priori (black dash line, Figure 11) in the collocated CrIS CO profile. Compared to the CrIS retrieval using the TROPESS $x_a$ (Figure 10), the retrieval using the MOPITT $x_a$ resulted in a different CO profile, especially near the surface. This is due to dominant contributions from the a priori near the surface where the averaging kernels has lower sensitivity to the true profile.

**RC2**: 'Comment on egusphere-2023-1369', Anonymous Referee #1, 20 Oct 2023

Review of "TROPESS CrIS CO single pixel vertical profiles: Intercomparisons with MOPITT and model comparisons for 2020 US Western wildfires" by Luo et al.

This paper evaluates the TROPESS Level 2 CO product against another well-known satellite product (MOPITT CO) as well as a climate model (GISS). Hyperspectral infrared instruments, like CrIS and AIRS, reliably measure CO concentrations in the mid-troposphere. This is especially useful when the plumes from large fires enter the free troposphere and get transported across continents with prevailing trade winds. With an average lifetime spanning multiple weeks, CO can be used as an air quality tracer as polluted air is transported from source to sink. TROPESS is a relatively new product so this paper adds useful information as to its quality and application. Moreover, this paper demonstrates how disparate products can be inter-compared in a scientifically meaningful way. The concerns I wish to highlight here are broadly centered around the fact that the conclusions appear unsupported by the analysis and results.

Thanks for the statements about the importance of Carbon Monoxide in air quality research.

Comments:

- This paper lacks reference to the NASA and NOAA operational Program-of-Record (PoR) for hyperspectral IR sounders. The AIRS L2 CO product is publicly available for more than two decades, the NOAA NUCAPS L2 CO product is available in near real-time from CrIS and IASI radiances and, the CLIMCAPS L2 CO product extends the AIRS record to CrIS on the JPSS series. It would benefit the community to see how TROPESS compares against these records. Can the authors explain this omission?

Some of the above-mentioned CO data products were briefly mentioned in the draft. But we indeed didn't list the complete references. We carefully added the  references/websites. We also added more recent satellite CO products suggested by the other reviewer. In addition, we briefly described TROPESS-AIRS CO products (Hegarty et al., 2022).

- The authors use multiple terms to refer to their product, which confuses the reader and muddies their arguments. I strongly suggest that they adopt and apply a consistent term; either TROPESS, MUSES or CrIS, but not all three. And I strongly recommend refraining from simply stating "CrIS CO" because the NOAA and NASA operational products for CO retrievals from CrIS radiance exist in the public domain with long lists of published resources. TROPESS CO is just another CO product from CrIS radiances. This should be clarified in the paper.

Thanks for the suggestion. We went through the paper draft end-to-end to make sure we consistently refer TROPESS-CO clearly as the data product for the subject of studies.

- "rms" should be capitalized.

Corrected.

- "a Priori" should not be capitalized like this.

Corrected.

- Before publication, I recommend the authors re-read their paper and restructure the many awkward sentences throughout. Per example, lines 184-186: "The model-satellite data comparisons and using their differences to evaluate key parameters used in the model computations have been widely used (...). Here we use CO data from Western US wildfire events occurred September 2020...".

Thanks, and we made updates to the identified awkward sentences.

- Overall, I'm concerned by the lack of sufficient citations, either to support the authors' statements, or to reflect the wealth of existing knowledge about satellite retrieval theory, products and practice. E.g., Line 210 reads "This retrieved profile obtained via applying the retrieval operator is the proper way of comparing the model to the satellite data retrievals." And I dare say, that the method the authors adopted for comparing models with satellite profiles is valid only within a limited experimental framework, as suggested by the many peer-reviewed studies already in print (but not cited). I suggest the authors clarify and define the value of their results within the scope of their experiment and the literature in general. And I strongly recommend that they refrain from using generalized, unsubstantiated statements such as the one highlighted here.

We tried to modify statements in the draft to avoid generalizing the specific inter-comparison and applications of TROPESS and MOPITT CO data described in the draft.

The retrieved profiles of an atmospheric constituent from satellite observations are strongly influenced by the choices of the a priori knowledge about the profiles. When one makes comparisons of the retrieved profile (not the true profile at the observation location and time) data to a model simulation of the given species or a species profile measured by in-situ instrument, one has to consider to make use of the retrieval operator provided by the retrieval teams (equation 1). These concepts are written in both TROPESS (https://docserver.gesdisc.eosdis.nasa.gov/public/project/TROPESS/User_Guides/TROPESS-AIRS-CrIS_CO_L2_Product_User_Guide_v1_2-22-21.pdf) and MOPITT (https://www2.acom.ucar.edu/sites/default/files/documents/v9_users_guide_20220203.pdf) documents with references, e.g., Rodgers 2000. There are indeed other ways combining atmospheric model and satellite observations, e.g., the assimilation process, but those serve a different purpose. The a priori data and the retrieval operator are still needed in these processes.

- The authors conclude with three statements; (i) a summary of the RMS results from a comparison between MOPITT and TROPESS CO retrievals, (ii) "…we illustrated the proper method for making model-satellite CO profile retrieval comparisons", and (iii) "We found the swapping the a priori mathematically works well."

We change the section title to "Summary". The method used in the analyses either for satellite tracer inter-comparison or the model-satellite CO profile comparison is not new. We have referenced some earlier works by us and others.

I have strong reservations about each statement, as follows:

(i)     The authors adopted a co-location range of 24 hours and 500 km between MOPITT and TROPESS CO profiles. This range can be considered reasonable when comparing inter-continental transport of pollutant air. Their focus, however, is a single wildfire event in 2020. Moreover, the authors quantified a bias with 23% RMS, 12-15% of which can be explained by "natural variabilities" within the space-time domains they used. But their study domain does not span enough time for this range of natural

variability to manifest. They are comparing two retrieval products during a single fire event. Overall, I am not convinced that their study is scientifically rigorous enough.

Thanks for pointing out the lacking of data comparisons in the draft. In the TROPESS-CO vs MOPITT CO comparison process, we have paired for more than one-day global data. Data shown in Figures 1-4 and the comparison statistical summary table 1 is for the global data Sept 12, 2020. The fire events are only in part of the US west coast. Like every day of a year, we see seasonally dependent CO enhancements in different places, e.g., tropical biomass burnings, megacity pollutants, boreal forest fires etc.

We added descriptions on the similar statistical results from the comparisons we've done for four representative days in the four seasons of the year. The comparison conclusion is about the same as for Sept 12, 2020 – compared to MOPITT, TROPESS CO is a few percent lower in the lower troposphere and slightly higher in the upper troposphere. We've also stated that the TROPESS CO vs aircraft in-situ measurement comparisons also showed similar statistical results.

(ii)     The method the authors used to compare model and satellite CO profiles is at least a few decades old and rigorously applied in many, published studies. Not only do the authors fail to cite these studies, they communicate their results as if they are contributing new information. Perhaps I am missing some of the finer nuances.  Can the authors explain in more detail why they've drawn this conclusion? And perhaps highlight where, exactly, their methods deviate from the status quo?

Using satellite data, either the direct radiance measurements or the retrieved atmospheric gas distributions to improve model developments is indeed a major application objective for the satellite observations. These works have indeed lasted a few decades, including weather forecast improvements, ozone trend analyses, and recently the air quality forecasts. It's perhaps the language used in the model-satellite comparison description (not a conclusion) that confused or gave a misleading impression on our intention of the 2nd point. We merely intended to extend the discussions of the principle used in satellite data inter-comparisons to illustrate a case of model-satellite comparisons. For the case of 2020 W US wildfire events, we illustrated that the GISS-E2 model tracer CO and the two satellite retrieved CO somewhat disagreed when the "retrieved" model profiles are compared to the satellite retrieved profiles.  Some detailed analyses illustrate that the raw model CO near the surface is large due to the fire CO emissions. However, the satellite retrieval sensitivities near the surface is quite low.  The retrieved max CO layer in the mid troposphere over the fires did not show up in the "retrieved" model CO profiles. This analysis posts a challenge to the model development. We have referenced previous studies  that investigate the model  transport parameters.  We went over the paper draft to modified some misleading languages.

(iii)    This statement appears to be an afterthought. It is not scientifically rigorous to say a method simply "works well". Can the authors qualify this statement with reference to the sections in their results that aided them in reaching this conclusion?

The last section of the paper is to illustrate whether a retrieved profile with a "swapped"  a priori profile agrees with the profile obtained with the standard a priori/initial guess used in the end-to-end retrieval process.  The "swapping" method is used in the two previous sections in satellite CO inter-comparisons and model-satellite comparisons. The agreement  illustrated in the 3rd section supports the simple "swapping" method.  We updated language in the section to improve the presentation.

The paper "TROPESS CrIS CO single pixel vertical profiles: Intercomparisons with MOPITT and model comparisons for 2020 US Western wildfires" by Ming Luo et al. presents retrievals of CO from CrIS using the TROPESS retrieval approach, and compares them to MOPITT Version 9 CO data and to simulations with the GISS climate model. A wild fire event serves as an example. They focus on the differences that are produced by using different a priori data, and on the proper way to compare satellite data influenced by a priori information to model results.

General comment:

The effects to be demonstrated and the methods used in this paper are not new. Nevertheless, the paper is a good illustration about proper use of satellite data in general, and specifically in comparisons to model data. However, I find several conclusions not being supported by the analyses that have been done. In particular, the sentence in the abstract "Comparison results that were adjusted to common a priori constraints in the retrieval processes have improved agreement between the two data sets over direct comparisons" is not supported by the results of the presented analyses. In my view, major revisions are necessary before the paper can be published.

Thanks for the evaluations of our manuscript. The methods described either for comparisons between the retrieved CO profiles from the two satellite observations by two project teams, or between model simulation and satellite retrievals are indeed not new. We've listed several published works by our authors and other groups, in which earlier works were also referenced. We realized it's probably the languages used in the draft that gave a misleading impression to readers about the originality of the comparison method. We edited the manuscript focusing on languages used.

Specific comments:

Abstract, l13/14: As already mentioned, I cannot see how this sentence is supported by the presented studies. See below for more details.

Introduction: The authors should make sure that comparisons to other data products of CrIS, possibly published earlier, are properly referenced, and shortly describe what the specific features of these data products, compared to the TROPESS-CrIS CO product are.

A lot of resources are needed in satellite gas profile retrieval comparisons between two projects. As suggested by other reviewers, we've added reviews of the satellite CO products and references, including some very recent data. As indicated in the title, the TROPESS species profile retrievals are done using single pixel observations.

l48-l66: A table comparing the relevant parameters of the two instruments and their retrieval approaches would be helpful.

Good suggestion. We added Table 1 for the observation configurations of CrIS and MOPITT.

l74/75: Is this supposed to be the natural variability without random errors of MOPITT? If so, how has it be derived?

The MOPITT daily (24 hour) CO retrieved profiles (they therefore include the MOPITT retrieval errors) are used in this analysis. The day is Sept 12, 2020.  We added "September" to the text. The variability was derived while we match TRPESS-CrIS and MOPITT COs within 500km for the day.

l77/78: Is the measurement error (noise in the spectral data) the only source of random errors considered? I.e. what else contributes to the precision of the data?

In the formal data error analyses, the measurement error or the precision for the retrieved CO profile is defined as the error due to radiance error estimated in Level 1B.

l79: "Different data processing teams use different a priori data." I am aware that this is the case. However, the true Bayesian approach would require that the a priori and the covariance matrix of the true known distribution of CO (i.e. derived from all available observations so far) was used. I hesitate to call approaches that follow formally the optimal estimation approach but use randomly designed a priori and S_a matrices to be Bayesian or "optimal estimation".

In theory, yes, the priori knowledge about CO global distribution at a given time and location should be the best available data. The decision is made by the science team of different project and documented in publications. These model data are evaluated by the in-situ and satellite data extensively.

l112: "is close to its "truth"""" I guess what is meant here is that linearisation is valid. You should say so if this is meant.

Yes, the linearization is valid. This is added.

l119:"Since TROPESS-CrIS retrievals use similar a priori CO profiles and constraints as MOPITT products,...": I do not agree to this statement, looking at Fig. 2 There are considerable differences in certain regions. If the two a prioris were indeed so close, why then study the impact of the a priori as  done in Section 5?

We mean "the a priori CO profiles derived from similar atmospheric model and the same constraints". The sentence is modified.

Figures 3 and 4: I would prefer a presentation of the two data sets as scatter plot; this would be by far more instructive.

Figure 1 is a straightforward way in visually comparing CO global distributions at the geo-locations of the two instruments. They both demonstrate major distribution hot CO regions and the features over land/ocean and N/S hemispheres. These features are in good

agreement between the two data sets. However, due to the differences of the exact observation locations and times of the two satellite orbit tracks, their values disagree at some locations, e.g, the profile pair described in the last section. The statistic analyses results presented in Table 2 are the global results generated from every paired TROPESS-CrIS and MOPITT CO volume mixing ratios, not the zonal averages, and the differences are computed in percent. The percentage differences avoided the domination by the enhanced CO data in some regions.

As suggested by the reviewer we tried the scattering plots for the pairing comparisons. The correlation coefficients are not high, e.g., 0.4 and 0.5 at 681hPa and 215 hPa for direct comparisons. They improved to 0.58 and 0.53 at the two pressure levels when comparing the Xa adjusted TROPESS-CrIS and the smoothed MOPITT CO. No obvious biases are seen. From these analyses, we draw the same conclusion as from Table 2 for the step-adjustment CO comparisons. Some discussions are added.

l128: Why this way around? Has MOPITT the wider AKs? The AKs and/or vertical resolution has not been mentioned so far. Is the width of the footprint of any relevance?

Good point. The AK in MOPITT should be wider that of CrIS. We examined the Degree of Freedom for Signal (DOFS) in the two datasets. The TROPESS-CrIS CO DOFS are slightly larger than that of MOPITT CO globally. The smoothing process was then applied to TROPESS-CrIS CO. We added discussions.

157/158: There seems to be something wrong with this sentence; I understand that there are differences in the vertical representation of the total columns. To illustrate this, a figure with the vertical profiles of CrIS and MOPITT would be helpful.

You probably mean the total column comparison in Table 1. Since there is not an exact match in location and time for a pair of the CrIS or MOPITT observations, a statistical analysis makes a better sense.

l169 and other: It is necessary to explain how the degradation of horizontal resolution has been done. Is it just assumed that e.g. the GFAS 0.1degx0.1deg value is valid for the larger model bin as well? Or have the emissions being reduced with the inverse ratio of the areas covered? The latter seems to be the case since the scales are different in the two panels of Fig. 5 (btw, this needs to be mentioned in the Figure caption). Please indicate whenever different horizontal resolution play a role, how they were handled.

GFAS emissions at $0.1^{\circ} \times 0.1^{\circ}$ have been re-binned to the coarser $2.0^{\circ} \times 2.5^{\circ}$ ModelE grid, conserving total CO emissions. This is now mentioned at L193. We have noted the different CO emissions scales between the two panels in Figure 5.

L183: Similar; it is not clear in the paper, if the values in the observations and in the model are understood as representative for a larger area (when going from "dots" to 2deg x 2.5 degree areas, or if a "dilution" due to the extension over a wider area has to be applied.

As best I understand this comment, I would say: "The coarse resolution ModelE2 CO profiles are indeed a 'diluted' version of the actual profiles, or rather, the profiles from a

higher-resolution model. We have clarified the sampling procedure for comparison against retrievals at L208, noting the compromise between the model and satellite resolutions.

l185-188: These two sentences are difficult to understand. Some rewording is necessary.

We have reworded these sentences to hopefully be more clear, and to situate our work alongside previous example studies.

l189/190: This kind of sentence belongs into the introduction.

Thanks for pointing that out. We have removed these sentences and added a phrase about horizontal gradients at L36 in the Introduction.

l191: This information should have been provided much earlier, in Section 2.

Yes, they are added in Section 2.

l196/197: "where the outstanding high CO VMRs are most likely closer to the emission sources – the burning area at the ground over land." This is definitely not valid for the large structure with strongly enhanced CO over the Pacific.

The emission sources of CO are "over land" in CO maps.  The high CO over E Pacific Ocean are most likely due to the combination of the transport and the retrieval smoothing effect in retrievals. Discussions are added.

Fig. 6 and all similar figures: Do you show the same values for the dots and for 1deg x1deg resolution? I.e. the observed values with their small footprint are considered representative for 1deg x 1deg (i.e. for about 110 x 110 km?). This must be clearly stated in the paper.

The "maps" were generated by 2-D averaging of CO VMR within 1degX1deg grids from the data shown in geolocation "dots" in the left panel of Fig. 6.  Figure caption is edited for clarification.

l206/207: Do you want to say that both the time difference of up to 24 hours and the mis-location of the footprints within the 1deg x 1deg area can contribute to the differences? How does this fit to the assumption, that a small CrIS footprint is representative for the 1deg x 1deg bin?

Here we compare the CO distributions of the two observations in their daily data. The discussion focused on broad CO features over the region of consideration. The 1degX1deg grid box-averages seems demonstrating the broad distribution nicely, as the side-by-side comparisons of the left and right panels in Fig 6 showed. Since the grid-box averaging is done, CO values the 1degX1deg boxes are not merely a single footprint values in the box.

l208: "The precision (measurement error due to noise) and the total retrieval error ...": What error sources are included for the precision? and what for the total retrieval error? These quantities need to be introduced and discussed, in comparison, in Section 2.

Modified to "(measurement error due to noise in spectral radiances)". We added references from the documentations of the two projects. The estimated retrieval errors and precisions are listed in Table 2 of Section 2.

l209: "The MOPITT retrievals ...": which ones? where?

The sentence is modified by adding "One of the reasons is that the MOPITT retrievals …"

l210: are the values now considered representative for 2deg x 2.5 deg?

For the model data, yes, the CO values are at a grid of 2X2.5 lat/lon.

l217: "the CO emission source distributions in the model and the near surface transport effects are expected.": do you mean "... we expect to see the CO source distributions and the near-surface transport?"

Changed to "the CO emission source distributions and the near surface transport effects are seen in the model simulation."

l226-228: I do not understand this sentence.

We have rephrased this sentence at L259 to hopefully be more clear.

Figs. 8 and 9: I am surprised and confused about the differences that result when GISS model output is treated with either the CrIS or the MOPITT retrieval operator. From Figs. 6 and 7 I had expected that the results should be more similar, since the observed "reality" in Figs. 6 and 7 is represented in a rather similar way in the products of the two retrievals. I do not understand where the considerable differences in Figs. 8 and 9, right panels, come from. In particular I am surprised, that the signatures in the CrIS-applied distributions (Fig. 8, right panels) are so weak.

The raw model CO distribution at 750hPa (Fig. 8 bottom left) has a similar pattern with the satellite CO at 450hPa (Fig. 6). However, the raw model CO at 450 hPa (Fig. 8 upper left) wasn't like that of satellites (Fig. 6). We understand that the model transport mechanisms and parameters determine its CO distribution, for example, a weak smock injection mechanism would not bring massive CO into mid-troposphere.  It's near surface CO distributions (e.g., @750 hPa) reflect the emission source locations and near surface horizontal transport strength. We also understand that the CrIS or MOPITT CO retrievals are mostly sensitive to the true CO distributions in the mid-troposphere, e.g., @450hPa. However, near the surface (750 hPa), the CO retrieval sensitivity are much lower compare to the mid-trop. The model CO profiles after applying the retrieval operators (its "retrievals") shown in Fig 8 and 9 right panels, then (1) kept their lack-of-CO signatures in the mid-trop (450 hPa), and (2) dominated by the retrieval a priori near the surface (750 hPa) which are pretty much the background CO climatology for the month. We believe these model "retrievals" are explainable rather than a coding error.

l246-248: This is only true IF the a priori and the S_a matrix is constructed from observed climatologies. From Fig. 2, this seems not to be the case, since the two a priori distributions

are different. This should be mentioned here (again). Second comment: depending on the way the retrieval is set up, the initial guess should not play a role for a fully converged retrieval; however, the a priori does. I assume you use the same climatology for initial guess and a priori. Nevertheless, the two quantities should not be mixed up.

We added Fig 2 as a reference describing the different a priori used for TROPESS-CrIS and MOPITT CO retrievals. The two teams made these decisions for their CO retrievals. Yes, the initial guesses are also the a priori for CO retrievals of the two teams. We added this point in the text.

l259/260: It must be mentioned that the CrIS and MOPITT profiles have a factor of almost 2 in their peaks. Again, this is not consisted with Figs. 6 and 7 and with the earlier statement that they differ by 3 to 5%.

As discussed in Section 2, the global statistic differences between the two COs are a couple to a few percent. For a specific fire event at a specific location though, it's difficult to identify a pair profiles having the same enhanced CO value mainly due to mismatch in time and location. We've stated criteria used to choose TROPESS-CrIS and MOPITT profile pair. We add the note that for this CO profile pair, TROPESS-CrIS CO VMR is 2X of that of MOPITT.

l262/63: "indicating the very weak CO plume transports vertically or a weaker CO emission in model setup near the surface." I don't understand what you intend to say here. You can see this in the model fields directly, no need to first apply the retrieval operator.

The discussion is to state the possible issues in model for its simulated CO profile over the fires. Yes, the "retrieved" model profile is not much different from the raw one – this can by predicted. For the consistency of the comparison method used in the paper, we showed both the raw and the "retrieved" model profile.

Figs. 10 and 11: Again, the response of CrIS is so much higher than that of MOPITT. This is in contradiction to Figs. 8 and 9.

The model CO values at the surface (dashed red) is large due to emission over the fire (~200ppb as shown in Figs 8 and 9). But it's not large enough in addition to the low injection height to bring CO to mid-troposphere.

l282-284: I do not understand this sentence.

The "swapped" method is the method described in Section 2. We added the note.

l300: I understand that for the final retrieved profile (black solid), the a priori was exchanged according to Rodgers, 2000, eq. 10.48. Is this correct? And what you want to show is that the resulting profile is close (ideally identical) to a profile retrieved with the other x_a and S_a. Correct?

This "exchange" of the a priori is the same as the 1st step for proper comparisons of the two satellite retrievals described in Section 2. We use the reference Luo et al. 2007b, where the

original reference is Rodgers, C. D., and B. J. Connor (2003), Intercomparisons of remote sounding instruments, *J. Geophys. Res.*, **108**(D3), 4116, doi:10.1029/2002JD002299.

l308/309: In the Abstract you say: "Comparison results that were adjusted to common a priori constraints in the retrieval processes have improved agreement between the two data sets over direct comparisons." And here you say: "A better agreement between the two satellite data sets is achieved at the last step. The slight biases of CrIS CO compared to MOPITT are about 5% in the lower troposphere and -3% in the upper troposphere." I really cannot see where these findings have been made. To draw this conclusion, you should have compared the CrIS profile with MOPITT x_a and S_a (the blue one in Fig. 13) with the MOPITT profile. This has not been done. From looking at the figures, I see that the differences are still around 250 ppbv, with CrIS almost twice as high in the peak as MOPITT, and differences are definitely not in the order of 3 to 5%.

The statement in the Abstract is mostly based on analyses in Section 2, the step-comparisons between matched TROPESS-CrIS and MOPITT CO globally. The 5% in lower-trop and -3% in upper-trop that we stated in these lines (308-309) are in in Section 6 Conclusions, which are also from these global analyses. The exercise we've done in Section 5 is that we tried to retrieve a new TROPESS-CrIS CO using the MOPITT x_a and S_a. Due to the mis-match of time and location, the TROPESS-CrIS and MOPITT CO are indeed 2X apart for the selected case, the new retrieval should not change this fact.

l320-322: What has been demonstrated in this work is that exchange of a priori after the retrieval is (almost) similar to a retrieval that uses the other a priori from the beginning. This is what Rodgers (2000) describes in his Eq. 10.48. However, using the MOPITT a priori does not bring CrIS closer to MOPITT. This has been demonstrated for CrIS alone, i.e. MOPITT should removed from this sentence.

Yes, you are right. The MOPITT profile shown in these figures is merely a visual reference, e.g., the left panel of Fig 13.

Minor comments:
Thanks for catching them.

l117/118: It should be "... in Rodgers (2000)."

Corrected.

Fig. 4: Typo "TROPESS" in figure caption.

Corrected.

l169: Use ModelE2 (and not ModelE) throughout the entire manuscript.

Corrected.

Section 4 and later: Some rewording is necessary. It would be good if one of the native

english coauthors could have a look.

Comments from English coauthors: it's in decent shape!

l192: ... taken on September ...

Corrected.

l205:  "...in the CO distributions ... high CO over the Pacific ..."

Corrected.

l208: 2X => twice

Corrected.

l216: remove "are shown"

Modify the single sentence to two.

l218: "exabit" => exhibit

Corrected.

l 290: since you called this algorithm "TROPESS" all the time, you should stay with "TROPESS" here as well.

Corrected.

F13 is of low representation quality, needs to be exchanged.

Replaced with the original figure.

---

## Author Response (AR2)

**Report #1**

Anonymous referee #2

Tha authors have addressed my previous questions and comments. I have two further comments:

(1) Unfortunately, Figures 1, 2, and 4 are still of low quality. They should be replaced with higher resolution version or vectorized version instead of raster if possible.

I have replaced Figures 1, 2 and 4 with higher resolution plots.

(2) For the title, "intercomparisons of" and "model comparisons" seem to be duplicated. Maybe considering changing "model comparisons" to "model simulations".

Thanks for the suggestion. We adopted "model simulations" in the title.

**Report #2**

Anonymous referee #1

Lines 30-34, Line 74: Does the journal accept website references like that? I suggest citing the seminal papers associated with each instrument instead. This is, after all, a scientific publication.

Thanks for the suggestion. We updated the manuscript to cite the original papers for the satellite instruments.

Line 52: The new title for Section 2.1 suggests that there are "two satellite observations", but I think they mean two sources of satellite observations.

We changed 2.1 to "CO retrievals from two satellite observations".

Line 94: "We selected a few days TROPESS"... add "of"

Corrected.

Line 137: "Sept 2020" should be written out

Corrected.

Line 140: "derived from similar atmospheric model" ... add "a"

Corrected.

etc.